# Innate Immunity and Sex: Distinct Inflammatory Profiles Associated with Murine Pain in Acute Synovitis

**DOI:** 10.3390/cells12141913

**Published:** 2023-07-22

**Authors:** Natália Valdrighi, Arjen B. Blom, Juliana P. Vago, Henk M. van Beuningen, Elly L. Vitters, Monique M. Helsen, Birgitte Walgreen, Onno J. Arntz, Marije I. Koenders, Peter M. van der Kraan, Esmeralda N. Blaney Davidson, Fons A. J. van de Loo

**Affiliations:** Experimental Rheumatology, Radboud University Medical Center, Geert Grooteplein Zuid 8, 6525 GA Nijmegen, The Netherlands; natalia.valdrighi@radboudumc.nl (N.V.); arjen.blom@radboudumc.nl (A.B.B.); juliana.vagodasilva@radboudumc.nl (J.P.V.); henk.vanbeuningen@radboudumc.nl (H.M.v.B.); elly.vitters@radboudumc.nl (E.L.V.); monique.helsen@radboudumc.nl (M.M.H.); birgitte.walgreen@radboudumc.nl (B.W.); onno.arntz@radboudumc.nl (O.J.A.); marije.koenders@radboudumc.nl (M.I.K.); peter.vanderkraan@radboudumc.nl (P.M.v.d.K.); esmeralda.blaneydavidson@radboudumc.nl (E.N.B.D.)

**Keywords:** inflammatory disease, arthritis, synovitis, pain, sex differences, innate immunity

## Abstract

Joint pain severity in arthritic diseases differs between sexes and is often more pronounced in women. This disparity is thought to stem from biological mechanisms, particularly innate immunity, yet the understanding of sex-specific differences in arthritic pain remains incomplete. This study aims to investigate these disparities using an innate immunity-driven inflammation model induced by intra-articular injections of Streptococcus Cell Wall fragments to mimic both acute and pre-sensitized joint conditions. Nociceptive behavior was evaluated via gait analysis and static weight-bearing, and inflammation was evaluated via joint histology and the synovial gene expression involved in immune response. Although acute inflammation and pain severity were comparable between sexes, distinct associations between synovial inflammatory gene expression and static nociceptive behavior emerged. These associations delineated sex-specific relationships with pain, highlighting differential gene interactions (*Il6* versus *Cybb* on day 1 and *Cyba/Gas6* versus *Nos2* on day 8) between sexes. In conclusion, our study found that, despite similar pain severity between sexes, the association of inflammatory synovial genes revealed sex-specific differences in the molecular inflammatory mechanisms underlying pain. These findings suggest a path towards more personalized treatment strategies for pain management in arthritis and other inflammatory joint diseases.

## 1. Introduction

Osteoarthritis (OA) and rheumatoid arthritis (RA) are characterized by joint inflammation and pain, leading to significant morbidity and disability [1]. Notably, the severity of joint pain in RA and OA differs between males and females. In particular, women tend to report more severe joint pain due to OA and RA compared to men [2,3]. In line with this, laboratory studies in humans have described sex-specific differences in sensitivity to noxious stimuli, suggesting that biological mechanisms underlie such differences [4,5]. This highlights the importance of understanding the biological underpinnings of these sex-dependent disparities in inflammatory joint diseases to manage pain. 

Synovitis, the inflammation of the synovial membrane, is a central feature of RA and the majority of OA cases; it contributes significantly to joint pain and damage [6,7]. The innate immune system plays a crucial role in the pathogenesis and progression of both diseases [8,9,10], as well as in the regulation of pain perception in affected joints [11,12]. The innate immune system comprises immune cells such as macrophages and neutrophils, which are strongly activated by pattern recognition receptors (PRRs) like Toll-like receptors (TLRs). This triggers a cascade of events, including the release of reactive species from oxidation–reduction (redox) reactions, cytokines, chemokines, and other inflammatory mediators and the activation of more immune cells, and as a result, this cascade culminates into a pain response. Furthermore, innate immunity can be trained/primed, and a second challenge will lead to an even more drastic inflammatory response [13].

Differences between sexes in the innate immune system have been observed due to hormonal influences, genetic factors, and epigenetic modifications (for a review, see [14,15]). For instance, hormones such as estrogens and androgens can both directly and indirectly impact the functionality and activation of innate immune cells such as macrophages and neutrophils. These hormonal influences can ultimately affect the production of both pro-inflammatory and anti-inflammatory cytokines [16]. 

Research has suggested these sex-specific variations extend to pain pathways as well, with immune-mediated divergences being apparent. For example, males have been found to primarily rely on TLR4 signaling [17,18,19], while females generally depend more on T cell-mediated responses [20,21]. Given that distinct components of the immune system dominate in males versus females, it is plausible that pain experiences may differ depending on the active elements of the immune system at a given time. This could mean that a primary immune response to a trigger may result in different pain outcomes compared to a secondary challenge with the same trigger, potentially even varying between sexes.

Such findings prompt the question of whether sex-specific differences in acute and pre-sensitized innate immune responses within the synovium may lead to disparities in joint pain experiences. 

To investigate whether differences in the innate immune response are differentially connected to pain outcomes between males and females, we used the Streptococcal Cell Wall (SCW)-induced arthritis model. This model is a valuable model for studying innate-immunity-driven synovitis as this model is dependent on TLR2 signaling [22], exhibiting no direct dependency on TLR4 [23] and T cell-mediated responses [24]. Second, the SCW-induced arthritis model is characterized by mild joint inflammation, marked by acute neutrophil exudation followed by macrophage infiltration. Thirdly, due to its self-limiting joint inflammation, this model allows one to study inflammation in acute and pre-sensitized joints [23,24]. 

In this study, we aimed to assess sex-specific differences in pain behavior and link this to potential differences in underlying synovial gene expression and inflammation. Pain was studied in both a static and dynamic setting using behavioral tests that evaluate different aspects of pain. Furthermore, we investigated whether there are differences in the inflammatory mediators associated with pain onset in the context of acute versus pre-sensitized inflammatory responses in the synovium. Specifically, we focused on how these responses relate to candidate genes implicated in innate immunity. 

## 2. Materials and Methods

### 2.1. Animals 

Twelve-week-old male and female C57BL/6J mice (Janvier Laboratories) were used in this study. Four or five animals were housed per cage and maintained under standard conditions, with unrestricted access to food and water. Environmental enrichment in the form of nesting material and readymade shelters (Igloo) (two per cage) were supplied. The animals were allocated randomly and blindly by an uninvolved operator. Due to obvious phenotypic differences between the sexes, blinding was not possible during the outcome measurement. Data analysis was performed blindly. All animal experiment and procedures adhered to guidelines approved by the Ethical Committee of Animal Research of Radboud University and by the Dutch Central Committee for Animal Experiments (DEC approval n° 2019-0001; AVD1030020197986), according to the European Union Directive (2010/63/EU) and Dutch regulations (EC2013-235). As the nature of our study was exploratory, the sample size was informed by previous unpublished research. We utilized 80 mice in total, with an even sex distribution (all for use in one experiment). 

### 2.2. SCW Model

SCW were produced as described previously [25]. Two serial synovial activations were elicited via intraarticular injections of 25 μg SCW into the right knee joint in 6 μl saline, one week apart. The first activation was in a naïve joint, and the second was in a pre-sensitized joint, allowing for a comparison between the differences in the acute versus pre-sensitized innate immunity. Injections were performed under isoflurane anesthesia and by a single operator. The mice were euthanized via cervical dislocation under the same anesthesia post-injection on either day 1 (n = 16 per sex) or day 8 (n = 16 per sex) of the experiment. Control groups were injected saline (day 8; n = 8 per sex). Animal welfare was consistently monitored throughout the experiment by both the research team and veterinary staff, including weight; food and water intake; and a general assessment of activity, panting, and fur condition. The animals did not reach the human endpoint during the experiment; however, 4 males were excluded due to fighting during the acclimatization period.

### 2.3. Experimental Design

Pain behavior was assessed using two different methods to compare mechanical pain, at movement or standing, between the males and females. Gait analysis was conducted first using the Catwalk XT (Noldus, Wageningen, The Netherlands), as the animals were allowed to walk freely, therefore facilitating more passive measurement. Using the incapacitance tester (Linton Instruments, Palgrave, UK), weight bearing in the affected joint in a static setting was then evaluated as previously described [26]. Special care was taken to ensure the reliability of these tests. Training started two weeks before model induction. During the first week, the mice went through a mock handling/training procedure to allow them to become familiar with the experimental settings. In the second week, three baseline measurements were taken on days −7, −3, and day 0. Before initiating the experiment, the mice were placed in the procedure room half an hour in advance. Researchers, timings, and the sequence of procedures were kept consistent throughout the study for standardization. Following model induction, pain behavior was recorded just before the weekly intra-articular injection and the subsequent day, including the endpoint on day 8. 

### 2.4. Pain Behavior Assessment (Catwalk and Incapacitance Tests)

Gait analysis was conducted using the Catwalk-XT system to study mechanical pain triggered by SCW injections and to identify any possible differences between males and females. Briefly, the mice were allowed to freely walk across a glass plate runway (internally illuminated with a green light and a high-speed camera underneath to capture the illuminated footprints). A successful trial consisted of three valid runs (60% maximal variation in run speed) per mouse. The trial values were calculated as the mean over the three runs. To measure mechanical pain behavior, footprints were identified and digitally analyzed. The evaluated parameters included print area, max contact max intensity, and single stance related to the arthritic joint (right hind). The print area assessed the size of the paw used to support the affected hind. The max contact max intensity, previously used to evaluate neuropathic pain in the chronic constriction injury model [27], is related to the maximum weight distributed within the evaluated paw. The single stance is the measurement of the stand time of one hind paw alone in the platform and has been previously used for gait analysis in pain models [26,28,29]. The gait modifications quantified by these parameters are similar to limping in humans.

To evaluate mechanical pain in a standing position on the hindlimbs, weight-bearing asymmetry was assessed with the incapacitance tester. The test consists of placing a mouse in a designated compartment that positions them on their rear legs. The right and left hind paws were positioned on two separate scales which measure the weight distribution on each leg. Stable readings were taken when the mouse was in a stable position for at least 4 s. Weight distribution was measured in five serial measurements per mouse. The percentage of weight on the right limb was calculated per experiment, followed by the calculation of the mean per individual. 

### 2.5. Isolation of the Knee Joints

On day 1 and day 8 after the induction of SCW arthritis, the animals were sacrificed histological or RNA analysis. For our histological analysis, the knee joints were isolated and treated with 4% formaldehyde, decalcified in formic acid, subsequently dehydrated, and finally encased in paraffin. Frontal slices were sectioned at 7 μm, mounted on coated slides, and stained with hematoxylin and eosin (HE). Inflammation was scored adopting an arbitrary scoring system (0–3). Analysis was executed blindly by two observers: 0, no influx; 1, mild cellularity; 2, higher cellularity; and 3, very high cellularity. For RNA isolation, synovium was isolated using a previously described standardized method [30]. Two 3 mm biopsies were extracted left and right adjacent to the patella and stored in liquid nitrogen until further isolation. 

### 2.6. Isolation of RNA (Synovium) and Subsequent qPCR

Gene expression levels were quantified using quantitative real-time PCR (qRT-PCR). Synovium biopsies for RNA isolation were homogenized in TRIzol reagent (Sigma Aldrich, Amsterdam, The Netherlands) using the MagNA Lyser Instrument (Roche, Mannheim, Germany). Total RNA was isolated according to the manufacturer’s protocol. The RNA concentration was evaluated with the Nanodrop spectrophotometer and reverse-transcribed into cDNA. Specific primers (Table 1) and the SYBR Green Master Mix were used for the qRT-PCR reactions using the QuantStudio 1 Real-Time PCR System (Thermo Fisher, Singapore, Singapore). The reactions were represented as delta delta threshold cycle Ct (ΔΔCt) values and calculated as the difference between the negative delta threshold cycle (−ΔΔCt) of the genes of interest measured in the SCW synovium and the control (saline) synovium. The −ΔΔCt of genes of interest was calculated by subtracting the reference gene GAPDH. 

### 2.7. Association between Inflammation and Nociceptive Profile 

To present an overview of the link between inflammation and the extent of pain, we utilized spider charts based on Spearman coefficient values (r_s_). Since pain behavior is characterized by a negative value, we used the difference from the baseline, set as 0, thus yielding a positive value. Genes associated with inflammation that showed a proportional association with pain are represented by a positive coefficient and located closer to the outer circle of the chart. Conversely, negative correlations, which indicate an inversely proportional association between synovial gene expression and pain behavior, are situated closer to the center of the circle. Associations were sorted using the strength of association with pain at day 8 in the male group as a reference. The strength of association was determined as follows: a Spearman coefficient ranging from ± 0.38 to 0.50 signifies a moderate correlation, and a coefficient ranging from ± 0.51 to 1.00 indicates a strong correlation.

### 2.8. Statistical Analyses

Results are shown as mean ± SEM unless specified otherwise. The normality of all data was determined using the Shapiro–Wilk test. Pain behavior was examined through repeated measures two-way ANOVA, followed by Šídák’s post-test to facilitate a comparison between the sexes, controls, and the baseline. Male and female differences in macroscopic inflammation and gene expression were analyzed using two-way ANOVA, followed by Šídák’s multiple comparisons test. The P value was adjusted for multiple comparisons. Prism 6 software (GraphPad software, San Diego, CA, USA) was used for all statistical analyses except for the correlation tests, which were analyzed via Spearman correlation analysis using SPSS (IBM Corp., Armonk, NY, USA). The threshold for statistical significance was set as *p* < 0.05. 

## 3. Results

### 3.1. Joint Inflammation Induced by Serial Injections of SCW Assessed by Histology 

As joint pain and inflammation are often intertwined, we first assessed the extent of cell influx in the synovium and joint cavity via histological analysis to investigate sex-specific processes. SCW injections into naïve knee joints induced an acute inflammatory cell influx, significantly different from the saline-injected joints. This inflammatory response was enhanced upon a second SCW injection. However, we found no sex-specific differences in inflammation in both the acute and pre-sensitized joints (Figure 1). 

Although our investigation did not reveal sex-specific differences in inflammation, we questioned whether there might be sex-specific variations in the pain behavior, leading us to assess nociceptive responses.

### 3.2. Pain Behavior of Mice with SCW-Induced Arthritis 

The SCW injections elicited observable changes in pain behavior on days 1 and 8. The extent was comparable across all investigated parameters between both time points and sexes (Figure 2). Gait parameters: maximum loaded weight (max contact max intensity—Figure 2a), paw support (print area—Figure 2b) and single stance (Figure 2c), and static weight-bearing asymmetry (incapacitance—Figure 2d) were measured in the affected paw, revealing the significant main effect of SCW injections (*p* < 0.0001). No main effects were observed for sex or interaction (sex*SCW), except for a sex-specific effect in the max contact max intensity (*p* = 0.0227). No differences between male and female mice were found during the entire test period at any of the measured time points for any of the investigated parameters (Figure 2).

With the knowledge that both sexes exhibited similar pain behavior, we wondered about the relationship between this behavior and inflammation. This prompted us to explore the correlation between these two variables.

### 3.3. Association between Pain Behavior and SCW-Induced Joint Inflammation

The correlation between pain behavior and the degree of joint inflammation in both sexes was evaluated using the Spearman correlation coefficient (r_s_) (Figure 3). During both the first and second SCW responses, maximum loaded weight in the affected paw (max contact max intensity), paw support (print area), and single stance showed similar positive correlations with cell influx in both sexes. 

Notable differences between sexes were observed in the static weight-bearing asymmetry. During the first SCW response, females demonstrated a proportional association (r_s_ = 0.577), suggesting that greater inflammation corresponded with more pain behavior. In contrast, males showed an inversely proportional relationship (r_s_ = −0.374) between the extent of pain and inflammation. Although the correlations were weaker during the second response, the associations reversed: females displayed an inversely proportional association (r_s_ = −0.160), while males exhibited a proportional relationship (r_s_ = 0.347). The correlations, despite not reaching statistical significance, yielded insights into the potentially sex-specific relationship between pain and inflammation.

Having noted intriguing patterns in pain behavior (weight-bearing asymmetry) and inflammation correlation, we sought to conduct further research into the molecular processes that may underpin these patterns. Hence, we turned our focus to the study of synovial inflammatory gene expressions.

### 3.4. Gene Expression of Synovial Inflammatory Factors

Our study analyzed the gene expression of several factors involved in the innate immune response within the synovium in response to SCW injections. We evaluated genes associated with pattern recognition receptors (*Tlr2* and *Tlr4*), alarmins (*S100a8* and *S100a9*), chemokines (*Ccl2* and *Cxcl1*), cytokines (*Il1b*, *Tnfa*, *Il6, Il10,* and *Tgfb1*), redox signaling (*Cyba*, *Cybb*-NOX2, *Ncf4*, and *Nos2*), a neurotrophic factor (*Ngf*), immune cell markers (*Emr1*-F4/80, *Cd86* and *Arginase1*), and a pro-resolving mediator (*Gas6*; for full names, see Appendix A).

We first assessed the mRNA expression levels of these genes in mice subjected to one or two intraarticular SCW injections, comparing the results to mice receiving saline injections (Table 1). We then examined sex-specific differences during the acute and pre-sensitized SCW-induced responses by calculating the changes in gene expression compared to saline (ddCt). The majority of the investigated genes showed a significant response after both SCW injections in both sexes (Table 2–main effect SCW, *p* < 0.0001). Compared to the same-sex saline group, gene expression levels increased following the first and second SCW injections. Exceptions included *Nos2*, *Ngf*, and *Gas6* in males and *Nos2* in females, which only increased significantly after the second SCW injection. In terms of sex-specifc differences during the first SCW response, females exhibited significantly a higher expression of *S100a8, S100a9, Ccl2, Il1b, Il6, Il10, Cyba, Ngf, Arginase1*, and *Gas6* compared to males. In contrast, males displayed a significantly higher expression of *Tlr4* during the first response (Figure 4a). During the second SCW response, females showed a significantly higher expression of *S100a8, S100a9, Il1b, Il6, Il10, Cyba*, and *Arginase1*, while males exhibited a significantly higher expression of *Tlr4* (Figure 4b).

When comparing the first and second responses within each sex, males exhibited a significant increase in *S100a9*, *Cyba*, *Cybb*, *Ncf4*, *Ngf,* and *Gas6* expression during the second response, while females displayed a significant increase in *S100a8*, *S100a9*, *Tnfa*, *Il6*, *Cyba*, *Ncf4*, *Emr1* (F4/80), *Cd86*, and *Gas6* expression during the second response. The magnitude of difference in gene expression between the first and second SCW injections was similar between the sexes.

The findings indicate that sex-specific differences exist in gene expression levels during both the acute and pre-sensitized SCW-induced arthritis responses, with more pronounced differences in naïve joints responding to SCW. Additionally, both sexes showed an increased expression of several genes during the second response (Table 2; spider graphs-Appendix A).

In conclusion, our results demonstrate significant sex-specific variations in the gene expression of synovial inflammatory factors in response to SCW-induced arthritis. Given these observed differences and the known association between inflammation and pain behavior (weight-bearing asymmetry), we find it compelling to investigate the associations between specific inflammatory genes and pain responses and whether these relationships differ between sexes.

### 3.5. Synovial Inflammatory Factors and Sex-Related Weight-Bearing Asymmetry in Acute SCW Arthritis

Analysis using Spearman correlation revealed a distinct association between pain behavior and innate immune factors’ gene expression in both acute and pre-sensitized synovial tissue responses. Sex-specific patterns were also observed (Figure 5).

In males, there was a less positive correlation between pain and the inflammatory genes in the acute response; however, *Il6* (r_s_ = 0.464), *Ngf* (r_s_ = 0.429) and *Cxcl1* (r_s_ = 0.393) showed a moderate positive association. A few inflammatory genes, including *Nos2* (r_s_= −0.714) and *Gas6* (r_s_ = −0.714), showed strong negative associations with pain. 

Following the first SCW challenge, females presented a stronger correlation between pain and a large number of inflammatory genes such as *Cybb* (r_s_ = 0.762) and *Ncf4* (r_s_= 0.683). Notably, *Il6*, which had a positive correlation with pain in males, demonstrated a negative association in females (r_s_ = −0.381). *Nos2* showed a strong negative association both in males and females (r_s_ = −0.643–Figure 5a).

The second SCW injection revealed a shift in the associations. The majority of the investigated inflammatory genes in males showed moderate to strong positive associations with *Gas6* (r_s_ = 0.929), showing a significant positive correlation with pain. Synovial *Cyba* (r_s_ = 0.857) and *Il6* (r_s_ = 0.821) also showed a positive correlation with pain behavior. In females, *Nos2* (r_s_ = 0.738) emerged as the top gene associated with pain. Other inflammatory genes such as *Tnfa* (r_s_ = 0.575), *Tlr4* (r_s_= 0.536), and *Emr1* (F4/80−r_s_ = 0.476) were also associated with pain behavior in females (Figure 5b).

Additionally, the synovial *Ngf* exhibited a sex-specific pattern of association with pain behavior, showing a positive correlation in males during the first (r_s_ = 0.429) and second response (r_s_ = 0.679), but no correlation was present in females. 

The evidence of sex-specific patterns in gene expression levels linked to pain behaviors further stresses the need for personalized approaches in the study and treatment of inflammatory joint diseases.

## 4. Discussion

The disparities between the sexes regarding joint pain, innate immunity, and pain pathways are considerable. However, to the best of our knowledge, this is the first study to compare and contrast sex-specific differences in acute versus pre-sensitized synovitis and pain. The intricate interaction between these variables prompts us to examine the link between inflammatory pain behavior and acute versus pre-sensitized response using the acute SCW-induced arthritis model. Our principal objective was to elucidate the sex-specific differences in synovial inflammation and pain behavior in the context of SCW-induced synovitis and their potential association with innate immunity-driven factors. 

Our findings revealed the absence of sex-specific differences in joint inflammation induced by serial SCW intraarticular injections. The initial SCW response, indicative of the acute reaction, exhibited a reduced arbitrary score when compared to the second injection, representing the pre-sensitized response (Figure 1). Nonetheless, pain behavior remained comparable between the first and second responses and across sexes during acute SCW synovitis, with significant SCW effects observed across several investigated parameters (Figure 2). The absence of sex-related differences in pain behavior in our study is consistent with publications in the existing literature, demonstrating that sex-specific disparities in pain behavior are not consistently evident in inflammatory pain models [31] unless specific molecular blocking strategies, such as gene knockout animals or pharmacological interventions [19] or direct injections of single compounds [17], are applied.

When investigating the correlations between pain behavior (static weight-bearing asymmetry) and joint inflammation (cell influx) during acute SCW arthritis, our results revealed interesting correlations (Figure 3). In females, a proportionate association was noted after the first SCW injection, suggesting a greater cellular influx corresponded with amplified pain behavior. Males, in contrast, exhibited an inverse relationship, with heightened pain associated with reduced cellular influx. These findings imply the potential involvement of distinct inflammatory pathways in pain between sexes, which led to an evaluation of synovial gene expression. 

To pinpoint the specific inflammatory pathways potentially contributing to these sex-related differences, we analyzed a set of synovial genes orchestrating the innate immunity response. We observed sex-specific differences in gene expression during both the first (Figure 4a) and second SCW-induced arthritis responses (Figure 4b), with females generally exhibiting a more pronounced expression of immune-related genes than males. This aligns with the hypothesis that females typically mount a more robust immune response than males, which is well-documented across various disease models [14]. Interestingly, both sexes showed an increased expression of several genes during the second response, reinforcing the involvement of trained/primed immunity in this arthritis model [13]. 

Further, we attempted to identify the distinct inflammatory pathways between sexes potentially involved in pain, investigating the association between synovial gene expression and pain behavior (static weight-bearing asymmetry). These associations emphasized that specific genes in the synovium are linked to joint pain in the SCW arthritis model, with distinct patterns observed between males and females (Figure 5). Different patterns were also observed between the first (Figure 5a) and second SCW injections (Figure 5b). Some of the key genes identified in this study include *Ngf*, *Il6*, *Gas6*, *Cyba*, *Cybb*, and *Nos2*, among others. 

Sex-specific differences in several of the key genes flagged in our study, such as *Ngf* [32,33,34], *Il6* [35,36,37,38], and *Gas6* [39,40] have been previously described. However, only within the last decade has sex been acknowledged as a significant biological variable that can impact innate immunity responses [41]. Historically, male and female animals were frequently used interchangeably in experimental studies without much attention being paid to the potential effects of sex-specific differences on disease processes. Our study has added to our understanding of this issue by demonstrating not only the existence of sex differences in response to SCW-induced arthritis but also the complexity of these differences, which manifest in unique inflammatory pathways and diverse interactions between synovial inflammation and pain behavior. Interestingly, our data suggest that, even in cases where pain differences are not immediately apparent, a more detailed approach may reveal them.

One of the key genes identified in our study is *Ngf*, a neurotrophic protein that plays a critical role in nociception modulation. Synovial *Ngf* showed a moderate to strong association with pain during both the first and second SCW responses in males, while it demonstrated no such association in females. This indicates a potential sex-dependent role of *Ngf* in pain responses. The current body of clinical trials focusing on anti-NGF reveals a potential sex bias. The studies included more female than male participants, yet most failed to conduct a sex-stratified analysis. Notably, only Dakin et al. [42] and Shoji et al. [43] have conducted such analyses. In the tanezumab (anti-NGF monoclonal antibody) clinical program, the syndrome of rapidly progressive OA was observed more frequently in women [44]. Therefore, it is clear that the sex-divergent role of *Ngf* warrants further investigation, not only to understand its differential role in pain response but also to explain the increased prevalence of rapidly progressive OA in women observed in anti-NGF clinical trials. Our study takes a step in this direction by investigating the correlation between various synovial genes, inflammation, and pain during SCW-induced arthritis, uncovering further intriguing sex differences.

During the first SCW response, males exhibited a negative correlation between inflammation (cell influx) and pain, suggesting that inflammatory factors might exert a protective effect on pain. In general, the majority of investigated synovial genes were negatively associated with pain in males. This could indicate that, after the first injection, the male inflammatory response can more effectively eliminate the SCW, limiting the inflammatory response. The strongest negative correlations were observed for *Nos2* and *Gas6*. Specifically, *Nos2* is an enzyme catalyzing nitric oxide production in response to inflammation, and it plays a role in modulating inflammation [45]. On the other hand, *Gas6* promotes efferocytosis, an essential step in resolving inflammation [46]. Conversely, *Il6* and *Ngf*, proinflammatory and pronociceptive factors, showed the highest positive correlations (albeit moderate). In females, a positive correlation was noted between cell influx and pain, with most synovial genes positively associated with pain. The strongest association was found for *Cybb* (NOX2), a redox signaling factor involved in host defense and redox signaling. In line with our findings, a study showed that female resident macrophages exhibit greater NOX2-mediated bacterial killing compared to their male counterparts [47]. Interestingly, *Nos2* was also the top negative association in females, which may suggest a similar effect of this gene on pain between sexes in the innate immunity response of naïve joints.

Similar to the cell influx associations, synovial genes associated with pain underwent more significant changes in males after the second SCW injection. In females, a weaker negative correlation between pain and cell influx was reflected in most of the investigated synovial genes, which either showed weak associations with pain or no association at all. The one exception was synovial *Nos2*, which now displayed a strong positive association. This might be due to the immune-challenged cells (most likely macrophages) in the joint playing a more dominant role in host defense and pain after the second injection than during the first SCW response [48]. In males, there was a trend of increased cell influx, which correlated with more pain after the second injection (Figure 3). In line with this finding, the association between synovial gene expression and pain shifted dramatically from the majority of genes being protective to almost all being positively associated with pain. The exception was *Nos2*, which, in the second injection in males, showed almost no association with pain. Interestingly, Gas6 shifted from having a protective role to becoming the top gene positively associated with pain. While this was unexpected, considering Gas6 usually induces anti-inflammatory signatures, recent studies have also revealed a proinflammatory role for Gas6 in males [49]. Pain in males also showed very strong associations with synovial *Cyba* and *Il6*. The *Cyba* gene encodes the regulatory subunit for NOX2 [50]. Interestingly, these sex-specific differences in NOX2 subunit association may be due to where these subunit genes are encoded (*Cyba* in autosome 16 in humans [51]; *Cybb* is encoded in the X-chromosome) [52].

Overall, one can speculate that distinct inflammatory pathways of innate immunity are involved in eliciting sex-differentiated pain responses in both naïve and pre-sensitized joints. In males, pathways related to inflammation resolution, indicative of an M2-like response [53], seem to dominate. On the first day, neutrophils are drawn to the joint cavity by *Cxcl1*. Given their short lifespan, these neutrophils subsequently undergo efferocytosis, most likely mediated by *Il6* [54]. During the second SCW response, on day 8, *Gas6*-mediated efferocytosis [46] becomes prominent, operating in tandem with *Il6*. Moreover, our observation of M2-like response dominance in males is supported by the role of the gene *Cyba*, which is essential for M2 differentiation [55]. Consistent with our findings, it has been reported that androgens appear to facilitate M2 polarization [56]. 

In contrast, females appear to display a dominant M1-like response [57], as suggested by the associations of pain with pro-inflammatory factors at all investigated timepoints. Further, an oxidative burst indicated by strong associations with *Cybb* on day 1 and *Nos2* on day 8 supports this hypothesis. Intriguingly, estradiol is known to promote M1 polarization, adding another layer of support to our observations [58].

It is important to clarify, that the strong correlations observed between certain genes and pain in mice do not imply that these genes or a combination of them directly cause pain. On the contrary, it suggests that pain can serve as a parameter to uncover sex-specific differences in the molecular pathways associated with joint inflammation-related pain. It is worth noting that these differences in inflammatory pathways were not observed at the extent of gene expression levels but only in their association with pain. This could mean that it may not be the quantity of inflammatory factors causing the differences, but rather how specific factors might shift one pathway or another. Moreover, these identified genes might act as representatives of a sex-divergent pain pathway. In other words, these genes could be crucial components in distinct biological processes in males and females that lead to different experiences of pain. This divergence could be driven by varied responses to inflammation, differences in hormonal influences, or other sex-specific physiological factors. Through a better understanding of these sex-divergent pain pathways, we could further elucidate the complex relationship between inflammation, pain, and sex. 

There are several limitations to our study that should be considered when interpreting the results. First, our assessment of pain behavior was primarily focused on weight bearing and gait analysis, possibly overlooking potential sex-specific differences in other aspects of pain such as mechanical or thermal hyperalgesia. Second, our investigation was restricted to the acute phase of SCW arthritis in 12-week-old mice. The age and disease chronicity of our mice model might not align fully with OA or RA patients, limiting our findings’ extrapolation to these conditions. Moreover, we are uncertain whether the observed sex-specific differences would persist in older mice or during a more prominent adaptive immune response phase. This highlights the need for future research to study these disparities in older animals and chronic arthritis models. Third, our inflammation scoring scale only measures total inflammation without differentiating potential sex-differentiated inflammatory cell types. To ascertain such differences, a detailed study on the various immune cells responsible for the differences in synovial inflammation between the sexes would be indispensable. Another limitation is our focus on the gene expression profiles of synovial factors without assessing the protein levels or functional activity of these factors. Furthermore, as this was an exploratory study, we acknowledge that the synovial gene selection was biased, and the next step should involve an unbiased approach such as transcriptomic or RNA sequencing, which may reveal underlying mechanisms. Further research should also validate our findings at the protein level and explore the functional roles of these genes in the context of sex-specific differences and challenged immunity in the SCW arthritis model. Moreover, incorporating the use of gonadectomized mice could provide valuable insights into hormone-mediated sex differences.

Our study provides valuable insights into the sex-specific differences in innate immunity and associated synovium inflammatory pathways within pain during acute SCW arthritis. These findings have several clinical implications: (1) Tailoring pain management strategies—a better understanding of the sex-specific differences in inflammatory pathways may enable the development of more personalized pain management approaches. For example, males may benefit more from therapies targeting the M2-related pathway, while females may require treatments focused on M1-related pathways. (2) Guiding the development of new therapies—understanding sex-specific differences in innate immunity and inflammatory pathways could inform the development of novel therapeutics, such as NGF-neutralizing antibodies, which may be more effective in addressing sex-specific pain mechanisms.

## 5. Conclusions

In conclusion, we confirmed that biological differences play a crucial role in sex-based pain disparities in inflammatory joint pain. We demonstrated that solely relying on expression levels is inadequate for drawing conclusions; instead, analyzing associations between immune factors is necessary to reveal these sex-based differences. Future studies employing an unbiased approach may help uncover the underlying mechanisms, which will subsequently require validation in human subjects. Our study underscores the importance of considering sex-specific differences in innate immune responses within the context of inflammatory joint diseases, as they contribute to disparities in joint pain severity. By deepening our understanding of the underlying mechanisms, including the roles of key genes and pre-sensitized immunity, we aspire to facilitate the development of sex-specific therapeutic strategies that effectively manage pain in arthritis and other inflammatory joint diseases for both male and female patients. This comprehensive approach has the potential to optimize pain relief and improve the overall quality of life of individuals afflicted by these conditions.

## Figures and Tables

**Figure 1 cells-12-01913-f001:**
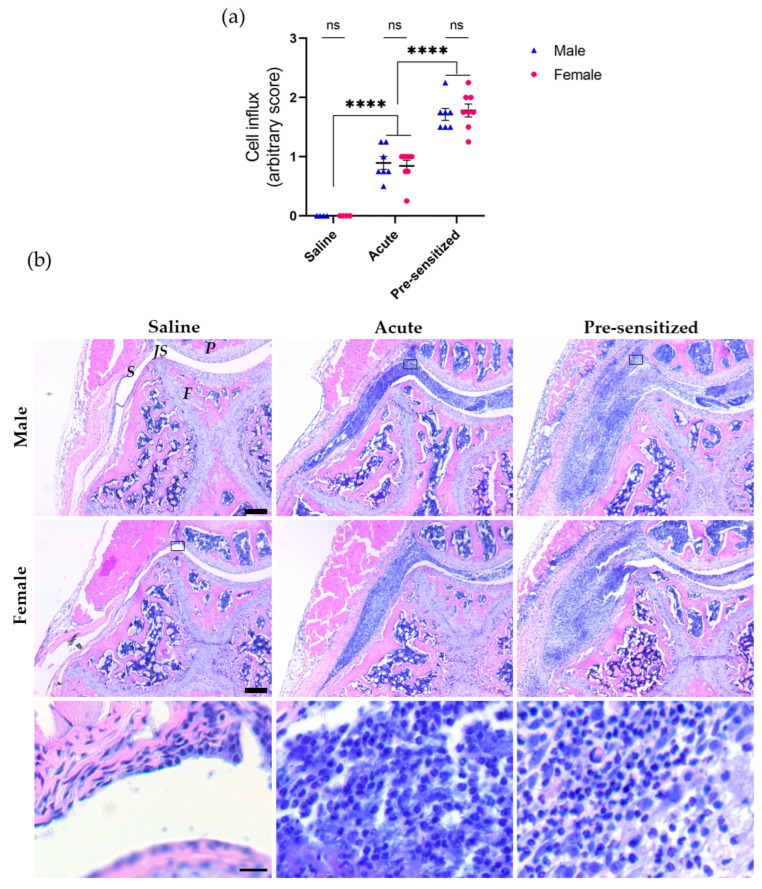
Joint inflammation of acute SCW arthritis showed greater cell influx in pre-sensitized joints but no sex-difference after the first or second SCW injection. (**a**) Microscopic inflammation was scored arbitrarily from 0 (meaning no inflammation) to 3 (corresponding to the maximum amount of inflammation). (**b**) Representative images of synovial inflammation in knee joint sections stained with hematoxylin and eosin (H&E) (scale bar = 100 μm) showing predominantly neutrophil extravasation into the joint cavity. The bottom three images show the selected areas at a higher magnification (scale bar = 10 μm). Two-way ANOVA with Šídák’s multiple comparisons test. **** *p* < 0.0001; ns: not significant. P = patella, JS = joint space, F = femur, and S = synovium. N = 7 males and 8 females per SCW group; N = 4 per saline group. Data are expressed as mean ± SEM.

**Figure 2 cells-12-01913-f002:**
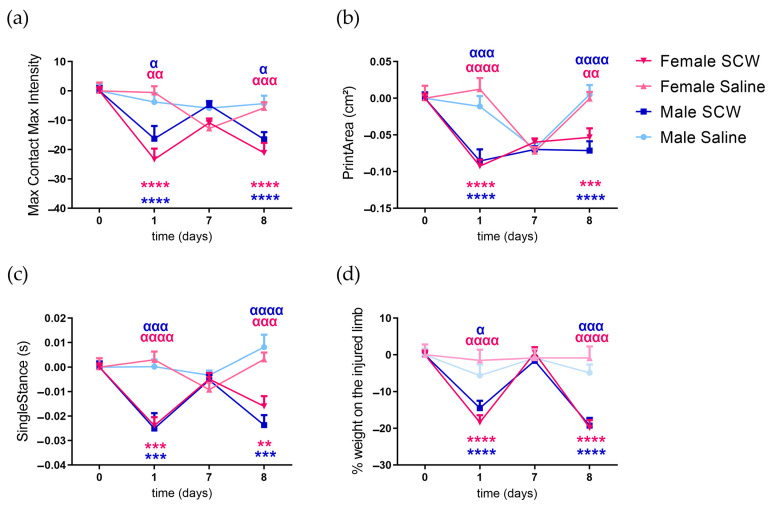
Pain behavior induced by SCW-induced arthritis was similar between sexes during acute and pre-sensitized responses. (**a**) Max contact max intensity, (**b**) print area, and (**c**) single stance on the right hind arthritic limb measured using the Catwalk system and (**d**) weight bearing on the right hind limb measured with the incapacitance tester. For all parameters, values are corrected for individual baseline, with negative values indicating pain behavior in the affected joint compared to baseline. RM two-way ANOVA and Šídák’s multiple comparisons test. * baseline comparison; α same sex saline comparison. α *p* < 0.05, αα or ** *p* < 0.01, ααα or *** *p* < 0.001, and αααα or **** *p* < 0.0001. N = 14 males and 16 females per SCW group; N = 8 per saline group. Data are expressed as mean ± SEM.

**Figure 3 cells-12-01913-f003:**
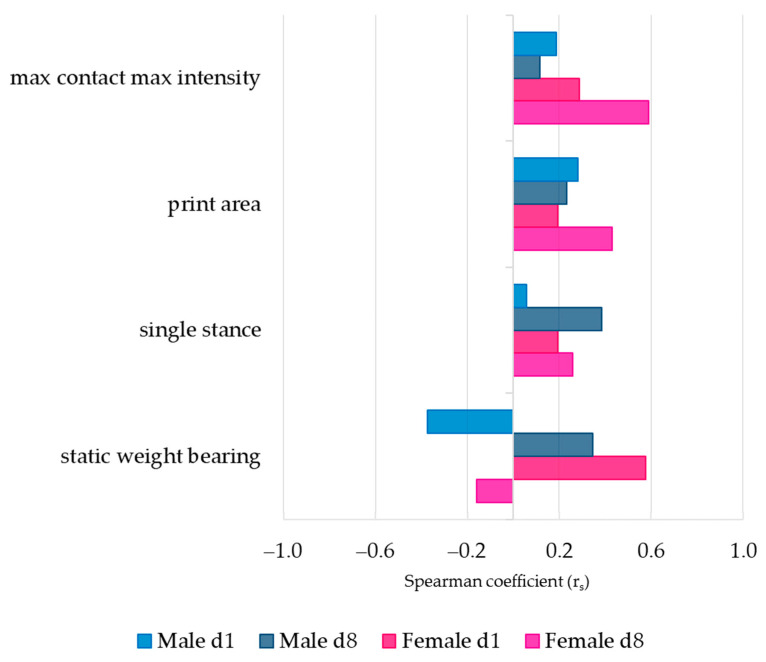
Correlation between SCW-induced joint pain and inflammation was weak and non-significant. Clustered bar graph based on the Spearman coefficient values (r_s_) between pain behavior parameters—max contact max intensity, print area, single stance on the affected (right) hind limb, and static weight-bearing (incapacitance)–and arbitrary score of inflammation. Coefficient values are displayed in the *x*-axis; inflammation with a proportional association with pain is indicated by a positive coefficient; r_s_ close to zero indicates no association between parameters; and negative correlations indicate a inversely proportional association between inflammation and pain behavior. N = 7 males and 8 females per SCW group. For specific r_s_ and *p* values, see Appendix A.

**Figure 4 cells-12-01913-f004:**
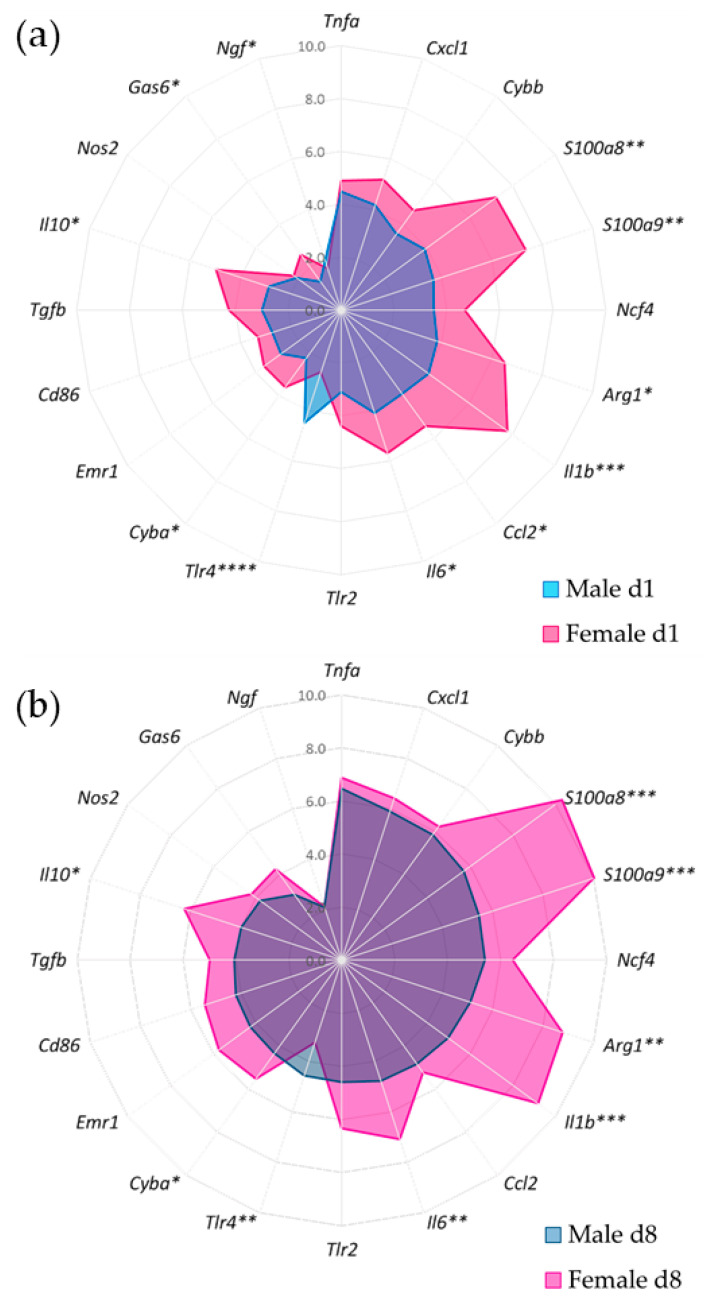
Synovial gene expression following intraarticular SCW injections showed a more robust inflammatory response in females. Spider chart based on the synovial genes expressed (ddCt) during acute SCW arthritis for the response after (**a**) the first SCW injection and (**b**) after the second SCW injection. Genes were sorted by stronger gene expression (ddCt) based on the male d8 group. Significance was calculated compared between sexes within the first or second SCW response using two-way ANOVA with Šídák’s multiple comparisons test. N = 7 males and 8 females per SCW group. * *p* < 0.05, ** *p* ≤ 0.01, *** *p* ≤ 0.001, and **** *p* ≤ 0.0001.

**Figure 5 cells-12-01913-f005:**
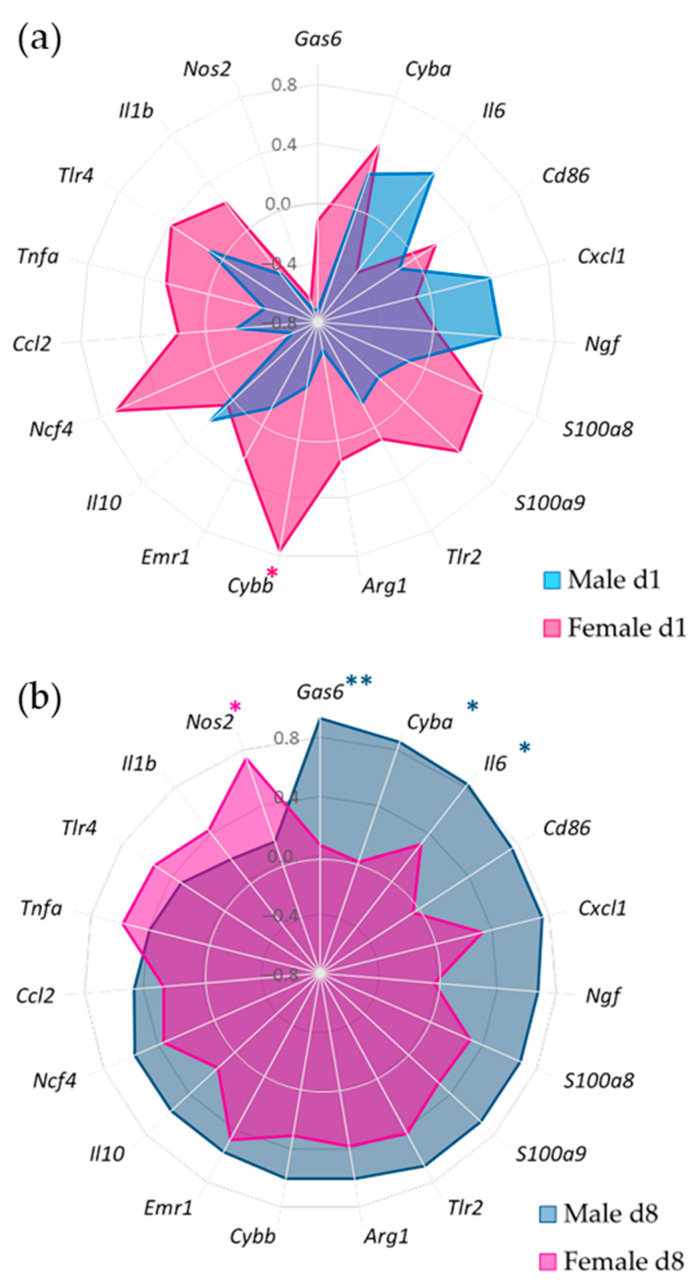
The correlation between synovial inflammatory factors and pain behavior exhibited different profiles between sexes. Spider chart based on the Spearman coefficient (r_s_) values between weight-bearing asymmetry and synovial genes expressed during acute SCW arthritis after (**a**) the first SCW injection and (**b**) after the second SCW injection. The genes were sorted by a stronger proportional association with pain based on the male d8 group. Inflammatory genes with a proportional association with pain are indicated by a positive coefficient and have been placed closer to the outer circle; the gray inner circle delineates r_s_ = 0, representing no association between parameters, while negative correlations indicate a inversely proportional association between synovium gene expression and pain behavior, located closer to the center of the circle. * *p* < 0.05 and ** *p* < 0.01. N = 7 males and 8 females per SCW group. For specific r_s_ and *p* values, see Appendix A.

**Table 1 cells-12-01913-t001:** Primers to measure gene expression in synovium using qPCR.

Gene	Full Name	Forward Primer	Reverse Primer
*Tlr2*	Toll-like receptor 2	5′-aacctcagacaaagcgtcaaatc-3′	5′-accaagatccagaagagccaaa-3′
*Tlr4*	Toll-like receptor 4	5′-ttccttcttcaaccaagaacatagatc-3′	5′-ttgtttcaatttcacacctggataa-3′
*S100a8*	S100 calcium-binding protein A8	5′-tgtcctcagtttgtgcagaatataaat-3′	5′-tttatcaccatcgcaaggaactc-3′
*S100a9*	S100 calcium-binding protein A9	5′-ggcaaaggctgtgggaagt-3′	5′-ccattgagtaagccattcccttta-3′
*Ccl2 (MCP-1)*	C-C motif chemokine 2	5′-ttggctcagccagatgca-3′	5′-cctactcattgggatcatcttgct-3′
*Cxcl1 (KC)*	C-x-C motif chemokine ligand 1	5′-tggctgggattcacctcaa-3′	5′-gagtgtggctatgacttcggttt-3′
*Il1b*	Interleukin-1 beta	5′-ggacagaatatcaaccaacaagtgata-3′	5′-gtgtgccgtctttcattacacag-3′
*Tnfa*	Tumor necrosis factor alfa	5′-cagaccctcacactcagatcatct-3′	5′-cctccacttggtggtttgcta-3′
*Il6*	Interleukin-6	5′-caagtcggaggcttaattacacatg-3′	5′-attgccattgcacaactcttttct-3′
*Il10*	Interleukin-10	5′-atttgaattccctgggtgagaa-3′	5′-acaccttggtcttggagcttattaa-3′
*Cyba*	Cytochrome b-245, alpha polypeptide	5′-gaggcaccatcaagcaacca-3′	5′-caccctcactcggcttcttt-3′
*Cybb (NOX2)*	Cytochrome b-245, beta polypeptide	5′-accctcctatgacttggaaatgg-3′	5′-cgaaccaacctctcacaaaggt-3′
*Ncf4*	Neutrophil cytosolic factor 4	5′-ccaactggctacgatgctactt-3′	5′-tctctggaactcacgcctcatg-3′
*Nos2*	Nitric oxide synthase, inducible	5′-cgtttcgggatctgaatgtga-3′	5′-gggcagcctgtgagacctt-3′
*Ngf*	Nerve growth factor	5′-tgcggccagtatagaaagct-3′	5′-ggggagcgcatcgagtttt-3′
*Emr1(F4/80)*	EGF module-containing mucin-like receptor	5′-actgtggaaagcaccatgttag-3′	5′-gctgccaagttaatggactca-3′
*Cd86*	Cluster of Differentiation 86	5′-agtgatcgccaacttcagtgaac-3′	5′-gcaggtcaaatttatgccagaat-3′
*Arginase1*	Arginase1	5′-gaaagttcccagatgtaccaggat-3′	5′-cgatgtctttggcagatatgca-3′
*Gas6*	Growth arrest-specific protein 6	5′-cctgccagaagtatcggtgatt-3′	5′-gtccaggattttcccgtttacc-3′
*Gapdh*	Glyceraldehyde-3-phosphate dehydrogenase	5′-cctgccagaagtatcggtgatt-3′	5′-gtccaggattttcccgtttacc-3′

**Table 2 cells-12-01913-t002:** Synovial gene expression one day after the first and second SCW intraarticular injections. ddCT ± SD * *p* < 0.05, ** *p* ≤ 0.01, *** *p* ≤ 0.001, and **** *p* ≤ 0.0001; ns: not significant. Significance was calculated compared to the same-sex saline group using two-way ANOVA with Šídák’s multiple comparisons test. N = 7 males and 8 females per SCW group; N = 4 per saline group.

		Male			Female		
		First Response	Second Response	1st × 2nd	First Response	Second Response	1st × 2nd
**Receptors**							
	*Tlr2*	3.1 ± 0.9 *	4.6 ± 1.5 ***	ns	4.4 ± 1.1 ***	6.3 ± 2.2 ****	ns
	*Tlr4*	4.5 ± 0.4 ****	4.6 ± 0.8 ****	ns	2.5 ± 0.8 ****	3.3 ± 0.3 ****	ns
**Alarmins**							
	*S100a8*	3.9 ± 1.4 *	5.7 ± 2.1 ***	ns	7.2 ± 1.5 ****	10.3 ± 2.2 ****	**
	*S100a9*	3.7 ± 1.4 *	5.5 ± 2.2 ***	*	7.3 ± 1.5 ****	10.0 ± 1.9 ****	*
**Chemokines**							
	*Ccl2 (MCP-1)*	3.9 ± 1.1 ***	4.8 ± 0.9 ****	ns	5.4 ± 0.9 ****	5.2 ± 0.8 ****	ns
	*Cxcl1 (KC)*	4.2 ± 1.5 ***	5.9 ± 1.5 ****	ns	5.2 ± 1.4 ****	6.4 ± 0.9 ****	ns
**Cytokines**							
	*Il1b*	4.1 ± 1.3 **	5.0 ± 2.3 ***	ns	7.8 ± 1.6 ****	9.2 ± 1.2 ****	ns
	*Tnfa*	4.5 ± 0.9 ***	6.5 ± 1.7 ****	ns	4.9 ± 1.4 ****	6.9 ± 1.2 ****	*
	*Il6*	4.1 ± 0.7 ****	4.8 ± 1.2 ****	ns	5.7 ± 1.0 ****	7.1 ± 1.1 ****	*
	*Il10*	2.9 ± 1.2 *	4.0 ± 1.0 **	ns	5.0 ± 1.6 ****	6.3 ± 1.3 ****	ns
**Redox signaling**							
	*Cybb (NOX2)*	3.6 ± 1.3 **	5.9 ± 1.7 ****	*	4.7 ± 1.1 ****	6.2 ± 0.7 ****	ns
	*Ncf4*	3.5 ± 1.2 **	5.4 ± 1.7 ****	*	4.7 ± 1.1 ****	6.5 ± 0.5 ****	*
	*Nos2*	2.1 ± 1.0 ns	3.8 ± 2.2 *	ns	2.2 ± 1.3 ns	4.2 ± 1.8 **	ns
**Neurotropic factor**							
	*Ngf*	0.6 ± 0.6 ns	2.1 ± 0.7 **	**	1.7 ± 0.6 **	2.1 ± 1.0 ***	ns
**Immune cells**							
	*Emr1(F4/80)*	2.8 ± 0.9 **	4.3 ± 1.4 ***	ns	3.6 ± 1.3 ***	5.7 ± 1.2 ****	**
	*Cd86*	2.7 ± 1.0 *	4.2 ± 1.7 **	ns	3.3 ± 1.2 **	5.4 ± 1.9 ***	*
	*Arginase1*	3.8 ± 1.9 *	5.1 ± 2.3 **	ns	6.5 ± 1.2 ****	8.8 ± 1.3 ****	ns
**Pro-resolution**							
	*Gas6*	1.3 ± 0.2 ns	3.0 ± 1.4 ***	**	2.6 ± 0.9 ***	4.3 ± 0.6 ****	**

## Data Availability

The data presented in this study are available in Appendix A.

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
