# Peer review of "Innate Immunity and Sex: Distinct Inflammatory Profiles Associated with Murine Pain in Acute Synovitis"

_cells, 2023, doi:10.3390/cells12141913_

Round 1
Reviewer 1 Report
Valdrighi et al investigated the immune mechanisms underlying sexual dimorphism in arthritis pain. Although no sex differences in joint inflammation and pain behavior were observed, they revealed that sex differences in innate immune profile. These findings are very interesting. However, low sensitivity of assays used in this study may weaken the reliability of data and interest in some experiments. I have major concerns with experimental design and conclusion.
Major concerns:
1) It is too arbitrary to draw no sex differences in pain behavior just based on weight bearing assay and gait analysis. How about other pain behaviors, such as mechanical hyperalgesia in the knee (primary) and hind paw (secondary) , and thermal hyperalgesia in the hindpaw?
2) HE staining may not be sensitive enough to detect the differences in joint inflammation between sex. Immunostaining specific immune cells (i.e. neutrophils, macrophages; T cells et al) in the joint sections or flowcytometry are recommended.
Minor concerns:
1) For SCW model, the dosage of SCW and the injection volume are needed in Methods part.
Reviewer 2 Report
This study found no sex differences in inflammation in both acute and pre-sensitized joints, but an association of inflammatory synovial genes revealed sex differences in the molecular inflammatory mechanisms underlying pain. Both sexes exhibited similar pain behavior, and sex differences exist in the gene expression during both the acute and pre-sensitized SCW-induced arthritis responses. The authors observed sex differences in gene expression during both the first and second SCW-induced arthritis responses, with females generally exhibiting more pronounced expression of immune-related genes than males.
These findings suggest a path towards more personalized treatment strategies for pain management in arthritis and other inflammatory joint diseases, with the development of sex-specific therapeutic strategies that effectively manage pain in arthritis and other inflammatory joint diseases for both male and female patients.
In my opinion, it is an excellent article. I have no comments to add.
Author Response
Dear Reviewer 2,
Thank you for your kind words! We appreciate the kind words and the interest
We sincerely appreciate your positive feedback and are pleased to know that you found our study valuable. It is indeed our goal to contribute to the development of personalized treatment strategies for arthritis and other inflammatory joint diseases. Your acknowledgment of our work's potential impact further motivates us to continue our research in this direction.
Thank you for your time and careful consideration of our manuscript.
Sincerely,
Natália Valdrighi
Reviewer 3 Report
In the present manuscript, the authors investigate the sex differences in pain behavior in a model of acute synovitis and correlate it with the histology of the injured knee, or the inflammatory response. The study is interesting and well-written.
The discussion is excessively long, and reducing its length may help the reader focus on the study's impact.
Major comments:
1. The authors chose to compare acute synovitis in 12-week-old mice. These mice are considered very young to assess models of osteoarthritis (OA) and may be limited for rheumatoid arthritis (RA). Hence, in the discussion, the author comments on the estradiol-driven M1 phenotype in female mice (Lines 511-513). The age of the mice does not represent OA announced in the introduction.
2. Building on the previous comment, a discussion on the use of ovariectomized mice to confirm the observed sex differences could be beneficial.
3. It is unclear how many times the in vivo experiment was repeated. Since the authors had 40 mice per group. Were some investigations repeated twice?
4. Although the author mentioned the need to investigate protein expression to complement the gene expression in their study, this investigation is crucial. Confirming the expression of a few cytokines/NGF would strengthen their findings and justify the thorough correlation with pain, histology, and behavioral tests, which are central to this article. The authors discuss the effect of anti-NGF antibodies in women (Lines 443-455), and it would be more relevant if they could point out differences at the protein level.
5. The interpretation of Nos2 gene expression needs further clarification since Nos2 is more likely involved in the progression of inflammation rather than its resolution.
6. In lines 304-308, it may be better to present the specific genes that differ between male and female mice, allowing the reader to check for differential regulation, as depicted in Figure 5a.
7. Did the authors perform any transcriptomic or RNA sequencing data analysis?"
Round 2
Reviewer 1 Report
The authors addressee my concerns.
Reviewer 3 Report
No additional comments